# Upregulated Expression of Activation-Induced Cytidine Deaminase in Ocular Adnexal Marginal Zone Lymphoma with IgG4-Positive Cells

**DOI:** 10.3390/ijms22084083

**Published:** 2021-04-15

**Authors:** Asami Nishikori, Yoshito Nishimura, Rei Shibata, Koh-ichi Ohshima, Yuka Gion, Tomoka Ikeda, Midori Filiz Nishimura, Tadashi Yoshino, Yasuharu Sato

**Affiliations:** 1Division of Pathophysiology, Okayama University Graduate School of Health Sciences, Okayama 700-8558, Japan; asami.kei@s.okayama-u.ac.jp (A.N.); gion@okayama-u.ac.jp (Y.G.); 2Department of General Medicine, Okayama University Hospital, Okayama 700-8558, Japan; 3Department of Medicine, John A. Burns School of Medicine, University of Hawai’i, Honolulu, HI 96813, USA; 4Department of Pathology, Okayama University Graduate School of Medicine, Dentistry, and Pharmaceutical Sciences, Okayama 700-8558, Japan; r.shibata.byouri@gmail.com (R.S.); me421004@gmail.com (T.I.); p2hq21br@s.okayama-u.ac.jp (M.F.N.); yoshino@md.okayama-u.ac.jp (T.Y.); 5Department of Ophthalmology, National Hospital Organization Okayama Medical Center, Okayama 701-1192, Japan; snowkohichi.1125@gmail.com

**Keywords:** activation-induced cytidine deaminase, IgG4-related disease, IgG4-positive marginal zone lymphoma

## Abstract

Immunoglobulin G4-related disease (IgG4-RD) is a systemic disorder characterized by tissue fibrosis and intense lymphoplasmacytic infiltration, causing progressive organ dysfunction. Activation-induced cytidine deaminase (AID), a deaminase normally expressed in activated B-cells in germinal centers, edits ribonucleotides to induce somatic hypermutation and class switching of immunoglobulin. While AID expression is strictly controlled under physiological conditions, chronic inflammation has been noted to induce its upregulation to propel oncogenesis. We examined AID expression in IgG4-related ophthalmic disease (IgG4-ROD; *n* = 16), marginal zone lymphoma with IgG4-positive cells (IgG4+ MZL; *n* = 11), and marginal zone lymphoma without IgG4-positive cells (IgG4- MZL; *n* = 12) of ocular adnexa using immunohistochemical staining. Immunohistochemistry revealed significantly higher AID-intensity index in IgG4-ROD and IgG4+ MZL than IgG4- MZL (*p* < 0.001 and = 0.001, respectively). The present results suggest that IgG4-RD has several specific causes of AID up-regulation in addition to inflammation, and AID may be a driver of oncogenesis in IgG4-ROD to IgG4+ MZL.

## 1. Introduction

It has been 20 years since Immunoglobulin (Ig) G4-related disease (IgG4-RD) was first described in 2001 [1]. It is a systemic disorder characterized by the formation of fibrotic lesions in various organs, including the pancreas, aorta, bile ducts, kidneys, and ocular adnexa [2,3]. Histologically, affected tissues of IgG4-RD exhibit dense fibrosis with infiltration of abundant lymphocytes, plasma cells, and eosinophils [2,4]. While the pathogenesis of IgG4-RD remains unclear; previous research noted that upregulation of cytokines secreted from T-helper 2 (Th2) and regulatory T (Treg) cell, including interleukin (IL) 4, IL-5, IL-10, IL-13, and transforming growth factor-beta 1 (TGF-β1) has been considered to play the crucial role in the onset of the disorder, causing the proliferation of IgG4- plasma cells [5].

Previous studies have argued that patients with IgG4-RD might have an increased risk of malignancies including colorectal, lung, and pancreatic cancer as well as malignant lymphoma than the general population [6]. In particular, some case reports have described the emergence of malignant lymphoma among patients with IgG4-RD [7,8,9,10].

Malignant lymphoma is classified into Hodgkin’s lymphoma and non-Hodgkin’s lymphoma. In Japan, B cell lymphoma accounts for about 68% of all non-Hodgkin lymphoma. Of these, diffuse large B-cell lymphoma is the most common (33.3%), followed by marginal zone lymphoma (MZL) (9.58%) [11]. Among all, ocular adnexal MZL has been noted as a unique entity that can arise in the context of IgG4-related ophthalmic disease (IgG4-ROD), including dacryoadenitis [12,13,14,15,16]. Indeed, it has been reported that 9% of cases with ocular adnexal MZL had significant infiltrates of IgG4-positive cells, suggesting a potential overlap of IgG4-ROD and MZL with IgG4-positive cells of ocular adnexa (IgG4+ MZL) [12]. According to a previous report in Japan [16], the histological classification of ocular adnexal lymphoproliferative disorders showed that 39.8% were MZL without IgG4-positive cells (IgG4- MZL), 21.6% were IgG4-ROD, and 4.3% were IgG4+ MZL. Male-to-female ratios of these conditions are almost 1:1, except for IgG4+ MZL, which occurs predominantly in males. We previously proposed that ocular adnexal IgG4+ MZL could arise in association with IgG4-ROD [17]. While some researchers tried to propose underground pathogenesis of IgG4+ MZL emergence in the setting of IgG4-ROD [18,19,20,21,22], the mechanism has been unclear to date.

Among molecules that could potentiate tumorigenesis, activation-induced cytidine deaminase (AID) is a member of the cytidine deaminase family enzymes that modify nucleotides causing various mutations. While AID plays a crucial role in B-cell tolerance, B-cell maturation, and antibody diversification by propelling somatic hypermutations in the Ig variable region to cause Ig class switching from IgM to other classes of Ig [23], AID expression can be found in various epithelial tissues due to chronic inflammation and infection [24]. Under normal circumstances, AID is expressed only in activated B-cells in germinal centers to prevent unnecessary mutations. However, there have been many reports of inappropriate AID expression in various cancers [24]. AID upregulation in chronic inflammation is believed to cause uncontrolled somatic mutations of multiple genes, resulting in the emergence of malignancy. Our group reported significantly more potent AID expression in extra-germinal centers in IgG4-RD than non-specific inflammation and control groups [25]. In this study, we aimed to clarify the extent of AID expression in ocular adnexal IgG4+ MZL and IgG4-ROD compared with MZL without IgG4-positive cells of ocular adnexa (IgG4- MZL) to see if AID plays a role in the pathogenesis of IgG4+ MZL while describing the histopathological characteristics of ocular adnexal IgG4+ and IgG4- MZL compared with those of IgG4-ROD.

## 2. Results

### 2.1. Clinicohistological Characteristics of the Patients

Table 1 summarizes basic demographics and histological characteristics of 16 patients with IgG4-ROD, 11 patients with ocular adnexal IgG4+ MZL, and 12 patients with ocular adnexal IgG4- MZL (for detailed data about each patient, refer to Appendix A). There were no significant differences in age between groups. Regarding the number of IgG4-positive cells, IgG4-ROD and IgG4+ MZL groups had significantly higher numbers of cells per high power field (/HPF) than those with IgG4- MZL. Likewise, the IgG4-positive cell/IgG-positive (IgG4+/IgG+) cell ratio was significantly higher in IgG4-ROD and IgG4+ MZL groups than the IgG4- MZL group. No statistically significant differences were noted in the number of IgG4-positive cells/HPF and IgG4+/IgG+ cell ratio between IgG4-ROD and IgG4+ MZL groups. Although no statistically significant differences were noted, IgG4-ROS and IgG4+ MZL groups had higher mean serum IgG4, IgG, and IgG4/IgG ratio than the IgG4- MZL group.

In patients with IgG4-ROD, all biopsy specimens showed marked lymphoplasmacytic infiltration, scattered eosinophils, and reactive lymphoid follicles. There were no light-chain restrictions. All cases fulfilled the criteria of IgG4+/IgG+ cell ratio being more than 40% (Figure 1).

In those with IgG4+ MZL, dense infiltration of medium-sized atypical lymphoid cells with a Ki-67 labeling index less than 10% were noted in all cases. These lymphoma cells were positive for CD20. Of note, abundant IgG4-positive cells with no light chain restriction and negative for CD20 were noted in all samples with the IgG4+/IgG+ cell ratio of more than 40% (Figure 2). Only scarce fibrosis or eosinophil infiltration were noted.

In IgG4- MZL, similar histopathological findings to those of IgG4+ MZL were noted except for the paucity of IgG4-positive cells (Figure 3). Except for three among 12 cases who had the IgG4+/IgG+ cell ratio of 49%, 47%, and 44%, the rest of them had a ratio of less than 40%. No cases had more than 50 IgG4-positive cells/HPF.

### 2.2. Immunohistochemical Analysis of AID Expression

Table 2 shows the result of immunohistochemical analysis of AID-expression. Immunohistochemical staining with the polyclonal antibody for AID revealed abundant AID-positive B cells in marginal zones of specimens from IgG4-ROD and IgG4+ MZL patients. Samples from IgG4- MZL consistently showed a small number of weakly AID-positive cells in marginal zones. The AID-intensity index to see the extent of AID expression was significantly higher in IgG4-ROD and IgG4+ MZL groups than in the IgG4- MZL group. There was no significant difference in the AID-intensity index between IgG4-ROD and IgG4+ MZL groups.

## 3. Discussion

This study examined the baseline demographics and histopathological characteristics of patients with IgG4-ROD, IgG4+ MZL, and IgG4- MZL, with additional focus on the extent of AID expression to clarify the pathogenesis of IgG4+ MZL. Ocular adnexa is the second most common site of extranodal marginal zone lymphomas (EMZLs) comprising 5–15% of non-Hodgkin lymphomas. While patients with EMZLs usually have indolent clinical courses, up to 50% of those with non-gastric EMZLs have been reported to experience disseminated diseases [26,27]. Thus, understanding of background pathophysiology of the disease is crucial for both clinicians and pathologists to allow for earlier interventions.

We previously reported that IgG4+ MZL of ocular adnexa might arise in a sequence of IgG4-ROD by immunohistochemical staining, quantification of IL-4, 5, 10, 13, TGFβ1, and FOXP3 using real-time polymerase chain reaction [17]. However, the underlying mechanism has remained unclear to date. One possible explanation is that IgG4 produced in tumor microenvironment and stroma may be associated with chronic antigen challenge associated with Th2 cytokines including IL-4, IL-10, and IL-13, which causes inappropriate host immune tolerance against malignancies and tumor progression [28]. Recently, IgG4 has been nominated as a critical molecule in the field of allergy and hypersensitivity [26] and oncology. As previous studies noted that the elevated levels of serum and tissue IgG4 were associated with worse prognosis in some types of solid tumors, IgG4 itself could be the cause of oncogenesis in IgG4+ MZL. Currently, it is still unclear whether tumor-specific or -nonspecific IgG4 plays a crucial role in cancer immune evasion and oncogenesis. Future studies are warranted to examine the role of tumor-specific IgG4 in IgG4+ MZL in ocular adnexa, using techniques including small interfering RNA (siRNA) or CRISPR/Cas9.

The other hypothesis is that AID overexpression may be a key finding propelling the oncogenesis. There has been growing evidence that AID overexpression has been noted to induce oncogenesis in some types of epithelial malignancies and oncogene translocation [24,29,30]. Simultaneously, it is a biologically crucial mutant enzyme that regulates class switch recombination of Ig. Therefore, AID deficiency might cause a complete defect in class switching and hyper-IgM syndrome. The present findings suggest that AID may play an important role in the oncogenesis of IgG4+ MZL in the setting of IgG4-ROD, given the significantly higher AID-intensity index in both IgG4-ROD and IgG4+ MZL than IgG4- MZL. It is worth noting that IgG4- marginal zone lymphomas predominantly express IgM, explaining the low AID-intensity index in the samples [31]. A study by Kasar et al. found AID activity even in indolent chronic lymphocytic leukemia, a disease entity known as one of hematologic malignancies with slower progression whole-genome sequencing [32]. Recently, efforts to reveal genomics and metabolomics profiles of IgG4-ROD and IgG4+ MZL of ocular adnexa are underway to reveal the underlying pathogenesis [21,22]. Given the current results, genomic level assessment and evaluation of AID activity may further help clarify the lymphomagenesis of IgG4+ MZL in the setting of IgG4-ROD. While AID has been suggested as a promising diagnostic marker as above, it also could be a therapeutic target. There is no established treatment for IgG4+ MZL, and similar treatment for IgG4- MZL, including radiation therapy for localized disease and systemic chemotherapy for advanced diseases, is mostly used. Tsai et al. reported that 5-aza-CdR, a DNA methyltransferase inhibitor, had an anticancer effect on AID+ hematopoietic cancer cells by downregulating AID [33]. Thus, a study to see the benefits of AID inhibitors like 5-aza-CdR is warranted in the future. As discussed, AID has a mutagenic activity that is related to oncogenesis. As noted in a study of double-hit lymphoma [34], targeting AID might be a good treatment regimen for IgG4+ MZL for which standard treatments have not yet been established [35].

Our study has a few limitations that need to be considered upon reviewing the result. First, the study was performed at a single Japanese university hospital, which reduces the generalizability of the results to patients with IgG4-ROD, IgG4+ MZL, and IgG4- MZL of ocular adnexa from other countries and facilities. In addition, given the disease’s rarity, we had difficulty collecting cases without missing data or specimens needed for the analysis. Thus, the small number of cases here needs to be noted. Secondly, given the scarcity of the orbital tissues available, we could not quantitatively measure the expression of AID messenger RNA and protein with PCR and Western blot, which warrants further investigations in the future. Additionally, given previous reports suggesting a possible role of follicular helper T cells and plasmacytoid dendritic cells in the pathogenesis of IgG4+ MZL, we would speculate the potential existence of these cells in MZL [36,37], although these need to be clarified in future studies. Despite the limitations, our study provides valuable data to elaborate upon the pathophysiology of oncogenesis of IgG4+ MZL of the ocular adnexa. AID could be a vital molecule to propel oncogenesis in patients with IgG4-ROD. The molecule may be a therapeutic target for IgG4-ROD to prevent potential progression to MZL

## 4. Materials and Methods

### 4.1. Samples

Formalin (10% formaldehyde)-fixed paraffin-embedded orbital tissues from patients with IgG4-related ophthalmic disease (16 cases), ocular adnexal IgG4+ MZL (11 cases), and IgG4- MZL (12 cases) were examined (Appendix A).

### 4.2. Histological Examination and Immunohistochemistry

All samples used in this study were surgically biopsied orbital tissue specimens (tissue in the cavity where eyes are situated). The procedures were performed at Okayama University Hospital, Mitoyo General Hospital, and Okayama Medical Center on different days. The specimen was fixed immediately in 10% formaldehyde after biopsy and embedded in paraffin, from which serial 3-μm-thick sections were prepared and stained with hematoxylin and eosin. The sections were then immunohistochemically stained with an automated BOND-III Stainer (Leica Biosystems, Wetzlar, Germany). Primary polyclonal antibodies against AID (ab59361, 1:150; Abcam, Cambridge, UK), IgG (A0423, polyclonal antibody, 1:20,000; DAKO, Glostrup, Denmark), and IgG4 (MC011, monoclonal antibody, 1:400; The Binding Site, Birmingham, UK), CD20 (L26 [1:400]; DAKO, Glostrup, Denmark), CD3 (LN10 [1:200]; Novocastra, Newcastle, UK), CD5 (4c7 [1:50]; Novocastra), CD10 (56C6 [100:1]; Novocastra), CyclinD1 (SP4 [1:50]; Nichirei, Tokyo, Japan), Ki-67 (MIB-1 [1:2500]; Novocastra) were used. In accordance with the consensus statement on the pathological features of IgG4-RD [26,27], three different high-power fields (HPFs) (total magnification, ×400) were examined to calculate the average number of IgG4+ cells per HPF and the IgG4+/IgG+ cell ratio. In situ hybridization was also performed for κ and λ -light chains (Leica Biosystems) using a Bond Max stainer.

### 4.3. Diagnosis of IgG4-ROD and IgG4+ MZL

To reduce the inter-rater bias, all cases were reviewed by two pathologists independently. Per the consensus statement for the pathological assessment of IgG4-RD [2,3] and the diagnostic criteria proposed by Goto et al. [38], different HPF (eyepiece, 10×; lens, 40×) were examined to obtain the average number of IgG4-positive cells per field (cutoff: 50/HPF) and the IgG4+/IgG+ cell ratio (cutoff: 40%). All 16 cases and 11 cases had histologically consistent findings as IgG4-ROD and IgG4+ MZL, respectively.

### 4.4. Analysis of the Extent of AID Expression

According to previous reports [2,3], we evaluated the extent of AID-expression based on an intensity index, with scores of 0 (negative), 1+ (weakly positive), 2+ (moderately positive), and 3+ (strongly positive) (Figure 4). Two pathologists independently evaluated each specimen.

### 4.5. Statistical Analysis

We analyzed the data using JMP version 15.1.0 (SAS Institute Inc., Cary, NC, USA). We used the Tukey–Kramer test to examine which pairwise comparisons are significant given the inequality of sample sizes between groups. The threshold for significance was defined as *p* < 0.05.

## Figures and Tables

**Figure 1 ijms-22-04083-f001:**
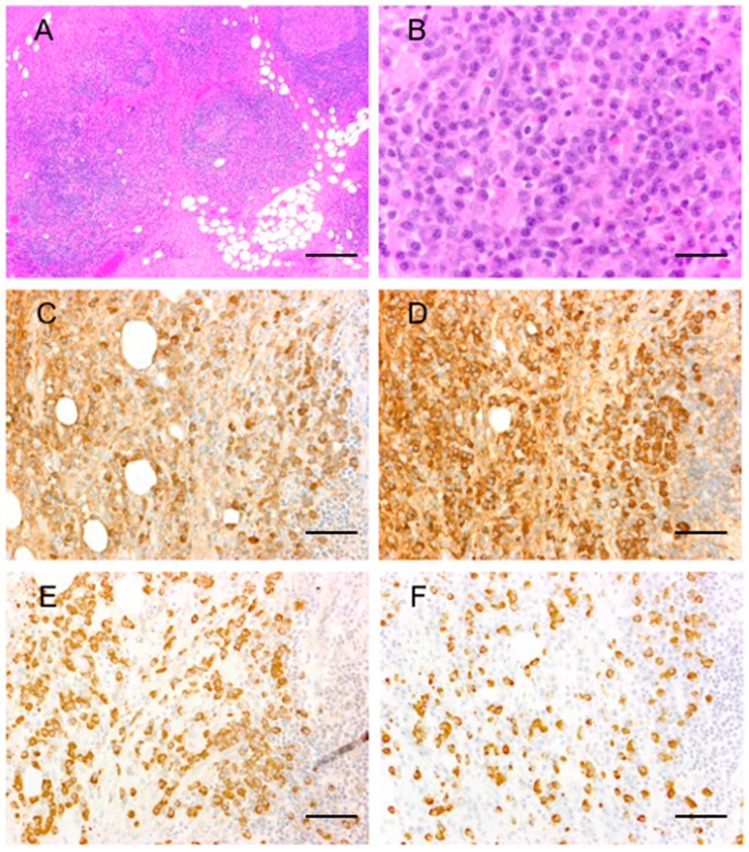
IgG4-related ophthalmic disease. Representative pathological images of IgG4-related ophthalmic disease. Marked lymphoplasmacytic infiltration with mild fibrosis, scattered eosinophils, and interspersed reactive lymphoid follicles (**A**,**B**) (hematoxylin and eosin). Numerous IgG+ (**C**) and IgG4+ (**D**) cells with an IgG4+/IgG+ cell ratio >40%. There were no light chain restrictions; Igκ-ISH (**E**) and Igλ-ISH (**F**). Scale bar; (**A**): 200 µm, (**B**): 50 µm, (**C**–**F**): 100 µm.

**Figure 2 ijms-22-04083-f002:**
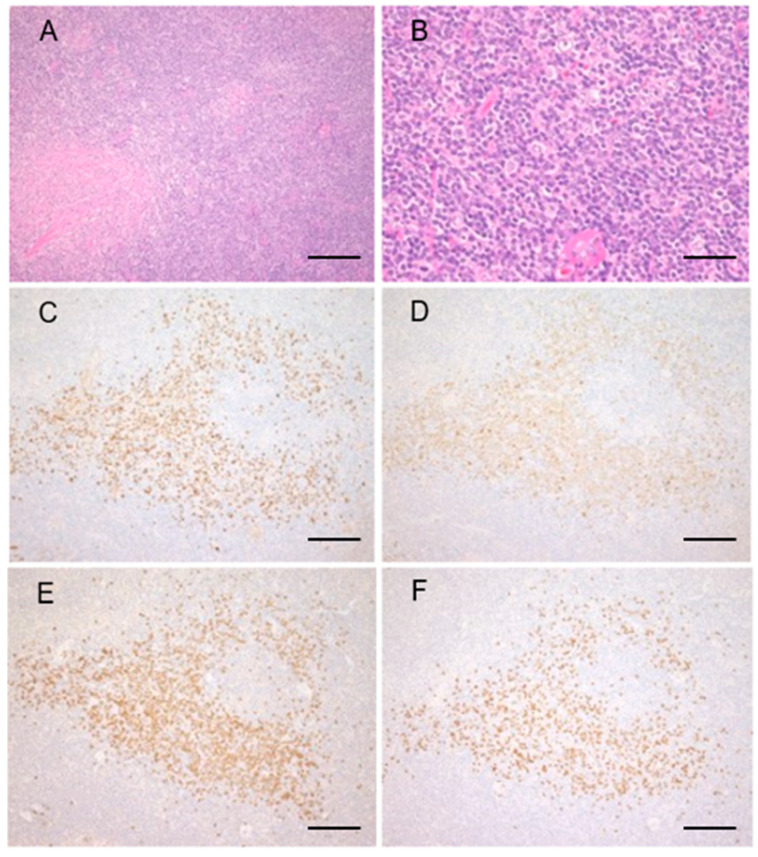
Marginal zone lymphoma with IgG4-positive cells. Representative pathological images of marginal zone lymphoma with IgG4-positive cells. The interfollicular expansion and diffusely proliferating atypical lymphoid cells with mild fibrosis (**A**,**B**) Infiltration of many IgG+ (**C**) and IgG4+ (**D**) cells are observed. The IgG4+/IgG+ cell ratio is >80%. Infiltrating B cells had no light chain restrictions; Igκ-ISH (**E**) and Igλ-ISH (**F**). Scale bar; (**A**): 200 µm, (**B**): 50µm, (**C**–**F**): 200 µm.

**Figure 3 ijms-22-04083-f003:**
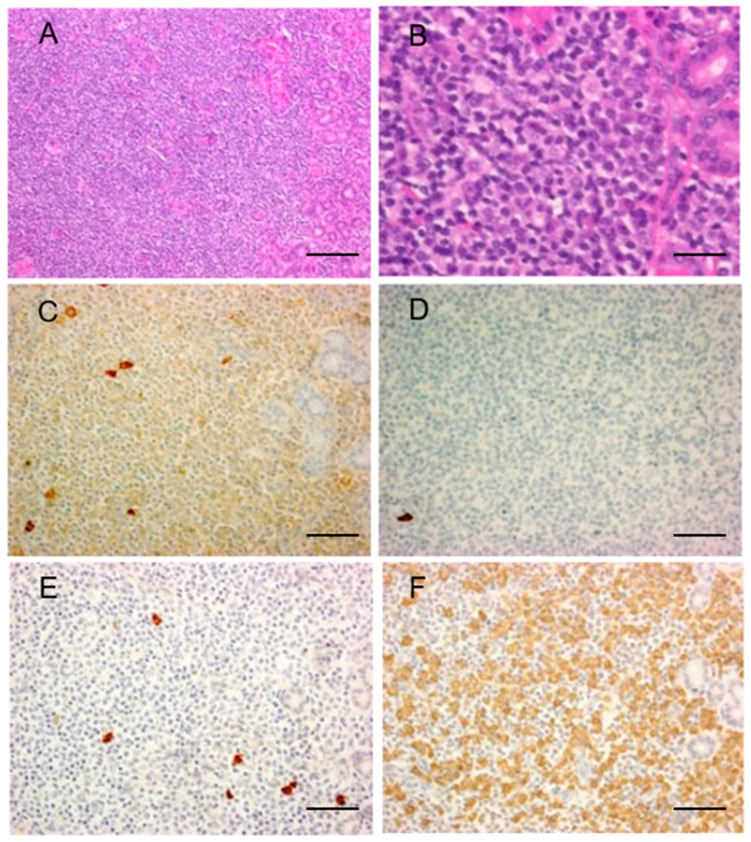
Marginal zone lymphoma without IgG4-positive cells. Representative pathological images of marginal zone lymphoma without IgG4-positive cells. Interfollicular expansion and diffuse proliferation of medium-sized atypical lymphoid cells (**A**,**B**). Low number of IgG+ (**C**) and IgG4+ (**D**) cells are present and the IgG4+/IgG+ cell ratio is <40%. A few Igκ+ B cells are present (**E**). Numerous Igλ+ B cells with light chain restriction were noted (**F**). Scale bar; (**A**): 200µm, (**B**): 50 µm, (**C**–**F**): 100 µm.

**Figure 4 ijms-22-04083-f004:**
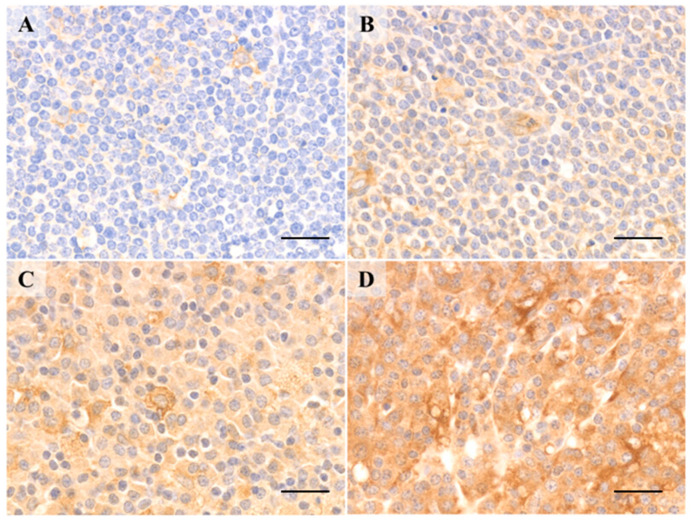
Activation-induced cytidine deaminase intensity index with immunohistochemical staining. Representative images of (**A**) Negative (0), (**B**) weakly positive (1), (**C**) moderately positive (2), and (**D**) strongly positive (3) activation-induced cytidine deaminase intensity index with immunohistochemical staining. Samples from patients with IgG4-related ophthalmic disease and IgG4-positive marginal zone lymphoma had strongly positive lymphoid and plasmacytoid cells with AID in both germinal centers and interfollicular regions (AID immunostaining, Scale bar; 50 µm).

**Table 1 ijms-22-04083-t001:** Basic demographics and clinicohistological characteristics.

	IgG4-ROD (Group 1: *n* = 16)		IgG4+ MZL (Group 2: *n* = 11)		IgG4−MZL (Group 3: *n* = 12)		*p*-Value		
	Prevalence (%)	Mean (SD)	Prevalence (%)	Mean (SD)	Prevalence (%)	Mean (SD)	Group 1 vs. 2	Group 1 vs. 3	Group 2 vs. 3
Age (years)		61.3 (3.1)		60.6 (3.7)		64.3 (3.5)	0.989	0.795	0.750
Sex									
Male	9/16 (56.2)		5/11 (45.4)		7/12 (58.3)				
Female	7/16 (43.8)		6/11 (54.6)		5/12 (41.7)				
IgG4+ cells (/HPF)		192.8 (90.2)		144.9 (88.0)		15.5 (20.3)	0.250	<0.001	<0.001
IgG+ cells (/HPF)		272.0 (124.4)		170.8 (73.3)		164.4 (234.1)	0.241	0.186	0.995
IgG4+/IgG+ cell ratio (%)		72.2 (5.6)		89.7 (6.8)		7.9 (6.5)	0.129	<0.001	<0.001
Serum IgG4 (mg/dL)		586.9 (192.0)		413.0 (221.7)		99.8 (332.5)	0.825	0.427	0.717
Serum IgG (mg/dL)		1704.0 (172.5)		1632.8 (215.2)		1377.6 (244.0)	0.964	0.527	0.716
Serum IgG4/IgG ratio (%)		24.7 (5.4)		24.7 (6.3)		6.8 (9.4)	1.000	0.247	0.272

Abbreviations: HPF, high power field; IgG4-ROD, IgG4-related ophthalmic disease; IgG4- MZL, non-IgG4-producing marginal zone lymphoma; IgG4+ MZL, IgG4-producing marginal zone lymphoma; SD, standard deviation.

**Table 2 ijms-22-04083-t002:** Immunohistochemical analysis of activation-induced cytidine deaminase expression.

	IgG4-ROD (Group 1: *n* = 16)	IgG4+ MZL (Group 2: *n* = 11)	IgG4- MZL (Group 3: *n* = 12)	*p*-Value		
	Mean (SD)	Mean (SD)	Mean (SD)	Group 1 vs. 2	Group 1 vs. 3	Group 2 vs. 3
AID-intensity index (0–3)	2.1 (0.20)	2.5 (0.24)	0.92 (0.23)	0.387	0.001	<0.001

Abbreviations: AID, activation-induced cytidine deaminase; IgG4-ROD, IgG4-related ophthalmic disease; IgG4- MZL, non-IgG4-producing marginal zone lymphoma; IgG4+ MZL, IgG4-producing marginal zone lymphoma; SD, standard deviation.

## Data Availability

The datasets generated and analyzed during the current study are available from the corresponding author on reasonable request.

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
