# Peer review of "Upregulated Expression of Activation-Induced Cytidine Deaminase in Ocular Adnexal Marginal Zone Lymphoma with IgG4-Positive Cells"

_ijms, 2021, doi:10.3390/ijms22084083_

Round 1

Reviewer 1 Report

Although the findings that upregulation of AID in ocular adnexal marginal zone lymphoma are interesting, a number of points need clarifying and certain statements require further justification. These are given below.

<Points>

1.        Concerning ethics, the authors described, “This study protocol was approved by the institutional review board of Okayama University (reference number 1607-016).”. However, the authors did not provide approval date. Please provide approval date.

2.        The authors showed little quantitative data. Please provide quantitative data such as ELISA, qRT-PCR, etc.

3.        Although the numbers of the disease are not large, the numbers of the study should be much larger.

4.        There is no scale bar in all figures (Fig, 1-4).

5.        In Acknowledgments, the sentence is incomplete.

6.        Ref. 33 is written in Japanese. Please replace English reference paper such as Goto, H. et al. (2015) Diagnostic criteria for IgG4-related ophthalmic disease. Jpn. J. Ophthalmol. 59, 1-7, 2015.

Author Response

Although the findings that upregulation of AID in ocular adnexal marginal zone lymphoma are interesting, a number of points need clarifying and certain statements require further justification. These are given below.

<Points>

  1. Concerning ethics, the authors described, “This study protocol was approved by the institutional review board of Okayama University (reference number 1607-016).”. However, the authors did not provide approval date. Please provide approval date.

[reply]

Thank you for your comments. We have added the approval data.

[line 271, page 9]

  1. The authors showed little quantitative data. Please provide quantitative data such as ELISA, qRT-PCR, etc.

[reply]

Thank you for your suggestion. While we acknowledge the importance of quantitative data, as mentioned, there were technical difficulties with qRT-PCR due to the small amount of orbital tissue samples available. We noted the importance of obtaining more quantitative data such as ELISA or qRT-PCR as limitations.

[line 203-206, page 7]

  1. Although the numbers of the disease are not large, the numbers of the study should be much larger.

[reply]

Thank you for your suggestion. Given the disease's rarity, we had extreme difficulty collecting cases without missing data/specimens for the analysis. We noted the small number of cases here as a limitation in the Discussion section.

[line 201-203, page 7]

  1. There is no scale bar in all figures (Fig, 1-4).

[reply]

Thank you for your suggestion. In the figures, we have noted the objective magnification in figure legends such as x40, instead of scale bars, following a common practice in the field of pathology.

  1. In Acknowledgments, the sentence is incomplete.

[reply]

We have completed the acknowledgements in the main text.

  1. Ref. 33 is written in Japanese. Please replace English reference paper such as Goto, H. et al. (2015) Diagnostic criteria for IgG4-related ophthalmic disease. Jpn. J. Ophthalmol. 59, 1-7, 2015.

[reply]

According to your suggestion, we have re-written the reference in English.

Reviewer 2 Report

This article shows that AID is increased in IgG4+ROD and IgG4+ MZL. As the disease is rare, the cohort here is important. Some points must be addressed: 

The authors present B cells are "plasma cells" in their manuscript despite they performed only CD20 staining. Moreover, Suimon Y et al (Anticancer Res 2020) showed that CD38 and CD138 were increased in IgG4+ROD compared to IgG4+ MZL. So, in IgG4+ MZL, they should use "B cells" instead of plasma cells.

Do patients presented recurrence of their lymphomas?

In IgG4- tumors, as IgG staining is low, what are Ig class? IgM? And if it is IgM, that could explained why AID expression is low in these samples.

As IgG4+ cells are found in MZL, are there also DC and Tfh in MZL? This point should be discussed. 

Author Response

<Points>

  1. The authors present B cells are "plasma cells" in their manuscript despite they performed only CD20 staining. Moreover, Suimon Y et al (Anticancer Res 2020) showed that CD38 and CD138 were increased in IgG4+ROD compared to IgG4+ MZL. So, in IgG4+ MZL, they should use "B cells" instead of plasma cells.

[reply]

Thank you for pointing this out. According to your suggestion, we have changed the word “plasma cells” to “B cells” where applicable throughout the manuscript.

  1. Do patients presented recurrence of their lymphomas?

[reply]

Thank you for your comment. No recurrence was noted in our cases during follow-up periods.

  1. In IgG4- tumors, as IgG staining is low, what are Ig class? IgM? And if it is IgM, that could explained why AID expression is low in these samples.

[reply]

Thank you for your suggestion.

A previous study reported that IgG4- MZL mainly expresses IgM. This may be the reason for the low expression of AID in these samples. We have added this point in the Discussion with additional reference.

[line 180-181, page 7]

  1. As IgG4+ cells are found in MZL, are there also DC and Tfh in MZL? This point should be discussed.

[reply]

Thank you for your suggestion. Given previous reports suggesting possible role of follicular helper T cells and plasmacytoid dendritic cells in the pathogenesis of IgG4+ MZL, we would speculate potential existence of these cells in MZL, although these need to be clarified in future studies. We have noted the point in Discussion with additional references.

 [line 206-208, page 7]

Reviewer 3 Report

This work used immunohistochemistry studies to investigate expression of AID in IgG4-related ophthalmic disease, marginal zone lymphoma with IgG4-positive cells, and marginal zone lymphoma without IgG4 positive cells of ocular adnexa. The study authors show significantly elevated AID-intensity index in IgG4-ROD and IgG4+ MZL compared to IgG4- MZL. The authors suggest that AID may b bring about oncogenesis in IgG4-ROD to IgG4+ MZL.

This is an interesting study.

There are some concerns.

The low sample size used in this study is a concern

Some information about lymphoma and types of lymphoma would be useful.

The authors need to give more information regarding ocular adnexal marginal zone B-cell lymphoma, incidence rate, information about how common the disease in the population, and other causes of the disease as well as how it is managed. Also, it would be useful to have more information regarding IgG4-related ophthalmic disease and marginal zone lymphoma. In addition, please provide male, female incidence ratio for these three conditions.

Some basic questions: Did the study authors perform the surgery to obtain the orbital samples and where (which hospital centre) was the surgery done? How much sample was obtained from each subject? Were all samples obtained in one day or on different days? Were the samples fixed in formalin at the site of surgery or brought to the laboratory and fixed? It will be useful to explain where the orbit is.

Was it 10% formaldehyde or 10% neutral buffered formalin that was used to fix the tissue? How much formaldehyde was actually present in the formalin solution.

What are the therapeutic options for IgG4+ MZL

A discussion regarding inhibitors of AID would be useful

A brief discussion of AID deficiency and implications would be useful

Minor comments

Line 21: induce somatic hypermutation

Line 161: that AID overexpression AID has been

Delete AID after the word overexpression

Et al needs to be in italics

Author Response

<Points>

  1. The low sample size used in this study is a concern

[reply]

Thank you for your suggestion. Given the disease's rarity, we had extreme difficulty collecting cases without missing data/specimens for the analysis. We noted the small number of cases here as a limitation in the Discussion section.

 [line 201-203, page 7]

  1. Some information about lymphoma and types of lymphoma would be useful.

[reply]

Thank you for your suggestion. We have added some background information about lymphoma in general in the Introduction.

[line 50-53, page 2]

  1. The authors need to give more information regarding ocular adnexal marginal zone B-cell lymphoma, incidence rate, information about how common the disease in the population, and other causes of the disease as well as how it is managed. Also, it would be useful to have more information regarding IgG4-related ophthalmic disease and marginal zone lymphoma. In addition, please provide male, female incidence ratio for these three conditions.

[reply]

Thank you for your suggestion. We have described the basic epidemiology of these conditions in Introduction.

[line 58-62, page 2]

  1. Some basic questions: Did the study authors perform the surgery to obtain the orbital samples and where (which hospital centre) was the surgery done? How much sample was obtained from each subject? Were all samples obtained in one day or on different days? Were the samples fixed in formalin at the site of surgery or brought to the laboratory and fixed? It will be useful to explain where the orbit is.

[reply]

Thank you for your suggestion.  The procedures were done in 3 different facilities on different days, and the samples were fixed immediately after resection in the OR. Since the volume or the orbit is generally considered to be around 30 ccs, the amount of samples taken were small. We have added some clarification regarding the points. 

[line 219-222, page 8]

  1. Was it 10% formaldehyde or 10% neutral buffered formalin that was used to fix the tissue? How much formaldehyde was actually present in the formalin solution.

[reply]

We used 10% formaldehyde for fixation. We noted this point in the main text.

[line 222, page 8]

  1. What are the therapeutic options for IgG4+ MZL

[reply]

Thank you for pointing this out. We have briefly described the current therapeutic options for IgG4+ MZL in Discussion.

[line 189-191, page 7]

  1. A discussion regarding inhibitors of AID would be useful

[reply]

Thank you for your suggestion. We have added discussion regarding 5-aza-CdR, DNA methyltransferase inhibitor, to downregulate AID, and its potential as a treatment for AID-positive hematopoietic cancers.

[line 191-197, page 7]

  1. A brief discussion of AID deficiency and implications would be useful

[reply]

Thank you for your suggestion. We have noted possible consequences of AID deficiency in the Discussion.

[line 176-177, page 7].

  1. Minor comments

Line 21: induce somatic hypermutation

Line 161: that AID overexpression AID has been

Delete AID after the word overexpression

Et al needs to be in italics

[reply]

According to your suggestion, we have revised the points

Round 2

Reviewer 1 Report

Judged by the revised version (IJMS-1157465-v2), a number of points that I previously pointed out remain in the revised version and certain statements require further justification. These are given below.

<Points>
1.        The authors did not show additional data in the revised version. Please provide additional quantitative data such as ELISA, qRT-PCR, etc.
2.        The authors did not provide an additional quantitative data in the revised version (IJMS-1157465-v2). This may be due to the small number of patients with this disease. If the number of patients with this disease is small, data should be accumulated until the number of patients increases a little, or the number of patients should be increased and analyzed in collaboration with other medical institutions.
Although the authors described the magnification, there is still no scale bar in all figures (Fig, 1-4). The photographs are usually magnified or reduced by printer. Therefore, the descriptions such as “Objective magnifications; A; x10, B; x40, C-F; x20” is no meaning. Please insert scale bar(s).

Author Response

Reviewer #1

Judged by the revised version (IJMS-1157465-v2), a number of points that I previously pointed out remain in the revised version and certain statements require further justification. These are given below.

<Points>

  1. The authors did not show additional data in the revised version. Please provide additional quantitative data such as ELISA, qRT-PCR, etc.

[reply]

Thank you for your suggestion. While we acknowledge the importance of quantitative data, as mentioned, there were technical difficulties with qRT-PCR due to the small amount of orbital tissue samples available. In addition, it was difficult to extract quality RNA from orbital tisssues which were small and fixed in formalin. We noted the importance of obtaining more quantitative data such as ELISA or qRT-PCR as limitations.

[line 203-206, page 7]

  1. The authors did not provide an additional quantitative data in the revised version (IJMS-1157465-v2). This may be due to the small number of patients with this disease. If the number of patients with this disease is small, data should be accumulated until the number of patients increases a little, or the number of patients should be increased and analyzed in collaboration with other medical institutions.

[reply]

Thank you for your suggestion. Although the number of cases of IgG4+ MZL and IgG4-ROD is small, our university is leading facility in diagnosis of those disease in Japan. If we collaborate with other facilities, the number of cases increases slightly, but we will require to get IRB approval again, and it is difficult to provide revised manuscript until deadline (within 7 days).

Although the authors described the magnification, there is still no scale bar in all figures (Fig, 1-4). The photographs are usually magnified or reduced by printer. Therefore, the descriptions such as “Objective magnifications; A; x10, B; x40, C-F; x20” is no meaning. Please insert scale bar(s).

[reply]

According your suggestion, we added scale bars in all figures.

[line 115-116, page 3, line 124, page 4, line 131, page 5]

Reviewer 2 Report

The authors answered to all points and adequately modified the manuscript.

Author Response

Thank you for your kind review.